# Non-Medical Factors Associated with the Outcome of Treatment of Chronic Non-Malignant Pain: A Cross-Sectional Study

**DOI:** 10.3390/ijerph19052881

**Published:** 2022-03-01

**Authors:** Irena Kovačević, Višnja Majerić Kogler, Valentina Krikšić, Boris Ilić, Adriano Friganović, Štefanija Ozimec Vulinec, Jadranka Pavić, Milan Milošević, Petra Kovačević, Davorina Petek

**Affiliations:** 1Department of Nursing, University of Applied Health Sciences, Mlinarska 38, 10000 Zagreb, Croatia; valentina@domnius.hr (V.K.); boris.ilic@zvu.hr (B.I.); adriano@hdmsarist.hr (A.F.); stefanija.ozimec-vulinec@zvu.hr (Š.O.V.); jadranka.pavic@zvu.hr (J.P.); 2Department of Family Medicine, Faculty of Medicine, University of Ljubljana, Poljanski nasip 58, 1000 Ljubljana, Slovenia; davorina.petek@gmail.com; 3School of Medicine, University of Zagreb, Šalata 3, 10000 Zagreb, Croatia; vkogler1944@gmail.com; 4Institution for Home Healthcare Domnius, 10000 Zagreb, Croatia; 5Department of Anaesthesiology and Intensive Medicine, University Hospital Centre Zagreb, Kišpaticeva 12, 10000 Zagreb, Croatia; 6Andrija Štampar School of Public Health, University of Zagreb School of Medicine, 10000 Zagreb, Croatia; milan.milosevic@snz.hr; 7Department of Rheumatology, Physical and Rehabilitation Medicine, University Hospital Centre “Sestre Milosrdnice”, 10000 Zagreb, Croatia; petrakova13@gmail.com

**Keywords:** chronic non-malignant pain, quality of life, depression, social support, self-treatment

## Abstract

Background: Chronic pain is a global public health issue with increasing prevalence. Chronic pain causes sleep disorder, reactive anxiety, and depression, impairs the quality of life; it burdens the individual and society as a whole. The aim of this study was to examine non-medical factors related to the outcome of the treatment of chronic non-malignant pain. Methods: A cross-sectional study with two groups of patients was conducted using a questionnaire with biological, psychological, and social characteristics of patients. Since this study was cross-sectional, it was not possible to determine whether some factors were the cause or the consequence of unsuccessful treatment outcome, which is at the same time one of the disadvantages of cross-sectional studies. Results: The poor outcome of the treatment of chronic non-malignant pain in a multivariate binary logistic regression model was statistically significantly associated with the lower quality of life (OR = 0.95 (95% CI: 0.91–0.99; *p* = 0.009), and higher depression level OR = 1.08 (95% CI: 1.02–1.14; *p* = 0.009). The outcome of the treatment was not directly related to social support measured by the multivariate binary logistic regression model (OR = 1.04, 95% CI: 0.95–1.15, *p* = 0.395), but solitary life (without partner) was (OR = 2.16 (95% CI: 1.03–4.53; *p* = 0.043). Conclusion: The typical patient with a poor pain management outcome is retired, presents depressive behavior; their pain disturbs general activity and sleeping. Moreover, they have a physically disturbed quality of life and require self-treatment due to the inaccessibility of doctors and therapies. The principle of treatment of patients with chronic, non-malignant pain should take into account a biopsychosocial approach with individually adjusted procedures.

## 1. Introduction

Chronic non-malignant pain (CNMP) has been recognized as a great public health problem that imposes significant economic and social burdens [1]; it has a prevalence prevalence in the population of around 20–50% [2]. European data indicate that moderate- and high-intensity chronic pain, with serious impacts on daily living, social status and working life, occurs in 19% of the European adult population [1].

Chronic non-malignant (CNMP) pain is defined with respect to its duration and recurrence. There are several definitions of CNMP [3,4,5]; one suggests that it is pain whose duration is longer than the time of expected withdrawal of the injury or completion of the surgery and lasts more than three months [3]. Chronic non-malignant pain (CNMP) reduces physical activity and may even cause disability which in turn affects other aspects of the patient’s everyday life [6].

Of many factors associated with chronic CNMP (depression, anxiety, excessive chronic pain awareness, coping strategies and pain belief), depression has been identified as the key factor. The higher intensity of pain is associated with depressive outcomes [7] and depression significantly affects effective pain treatment and influences the reduction in quality of life [7,8,9,10]. The musculoskeletal pain is one of the most common chronic, non-malignant conditions, and is strongly associated with depression [11]. Individuals who suffer from musculoskeletal pain, and in addition suffer with depression, are on sick leave for two times longer than those in pain but not depressed [12,13].

The results of the previous studies also indicate the association of chronic pain with various demographic characteristics. Some of the socio-demographic factors associated with chronic pain are female gender, old age, lower socioeconomic status, geographic and cultural background, employment status, and abuse and negative human relations in medical history [14]. Some studies reported the association of obesity with chronic pain in elderly men and women; consequently, moderate obesity was associated with a twofold higher probability for chronic pain occurrence in comparison to those of a normal body weight [15]. According to Shawe al. and Hestbaeket et al., there is also association of some factors with the site of pain, so that certain social factors, such as the income, accessibility to the health care [16] and lower educational level [17] may be related to the chronic back pain [18].

The pain includes all aspects of the patient’s nature, from the physiology and biochemistry, over-emotional and motivational appearance to psychological processes, social relationships and mental consciousness [19]. It was demonstrated that social support has an effect on chronic pain intensity, and various sources of support can encourage the person to use the facing strategy [20]. A literature review showed insufficient data in this field, especially in Croatian settings [21]. The aim of this study was to examine non-medical factors related to the outcome of the treatment of chronic non-malignant pain.

The objectives we wanted to achieve were to:Identify the differences in the quality of life, patient satisfaction, and self-rated health, in the group of participants with successful chronic non-malignant pain management, and in the group where the intensity of pain was not reduced despite treatment;Determine the differences in the biological, psychological, and social factors between the two groups;Establish the association of selected non-medical factors with the treatment outcome of chronic non-malignant pain.

## 2. Methods

### 2.1. Type of Research

A cross-sectional study with two groups of patients was carried out at the University Clinical Hospital Center “Sestre milosrdnice” in Zagreb (Croatia). The data were collected in a period of approximately1 and a half years (18 April 2018–15 November 2019).

### 2.2. Participants

The consecutive patients were visitors of the Department of Anesthesiology, Reanimatology, Intensive Medicine and Pain Management at the University Clinical Centre “Sestre milosrdnice” in Zagreb, aged 18 or older, with chronic, non-malignant pain. The patients gave informed consent for participation in the research. The study was approved by the Ethics Committee of the University Hospital Center (UHC) “Sestre milosrdnice” in Zagreb, Croatia, at its regular session held on 25 May 2016, file # EP-7811/16-6. The Ethical Committee of the University Hospital Center (UHC) “Sestre milosrdnice” operates in line with the principles of the International Conference on Harmonisation (ICH GCP) and the Helsinki Declaration. After a thorough written and oral explanation of the ethical principles, purpose, and course of research, they were asked to give informed consent. The first group of participants (156) were patients who had successful treatment of pain but were still visiting the pain clinic once per month, for following 4–5 months for a regular follow-up. The second group consisted of participants (180) who had been treated for pain in the Clinic for a longer period (more than a year) but did not experience significant pain relief. Every included patient from both groups answered a specially developed questionnaire. The nurse additionally checked whether the patients properly completed the questionnaire understood questions and signed informed consent to participate in the study. In this way, the number of non-answered questions was reduced, and the credibility of answers was increased.

### 2.3. Instrument

The research instrument was a survey of the factors that are associated with the treatment outcome of chronic non-malignant pain and additional questions about self-treatment based on the previously made qualitative study [22].

In this study, several internationally validated questionnaires/scales were used. All of them, except for the Patient Satisfaction Questionnaire Short Form (PSQ-18) [23,24], have already been validated in Croatia. For the PSQ-18, a 2-way translation and validation was performed.

The following questionnaires have been used:

Patient Satisfaction Questionnaire Short Form. Seven dimensions of patient satisfaction with physicians were assessed, including general satisfaction, technical quality, human relations, communication, financial aspects, time spent with the physician, and availability and convenience [23,24].

Functional Comorbidity Index (FCI) used to adjust for the effect of comorbidity on physical function. The FCI was specifically developed for use in a general population with physical function, not mortality, as the outcome of interest. The FCI as created by Groll et al. is an 18-item list of comorbid conditions without weighting (range 0–18) [25].

CES-D as a screening test for depression and depressive disorder. The 20-item scale remains 1 of the most widely used instruments in the field of psychiatric epidemiology. The score is the sum of the 20 questions. The possible range is 0–60. If more than 4 questions are miss answers, the CES-D questionnaire should not be scored. A score of 16 points or more is considered depressed [26].

Generalized Anxiety Disorder 7 (GAD-7) to measure the level of anxiety. In this form of self-assessment, subjects are asked to rate their anxiety-related problems on the 4-point scale (0 = not at all; 3 = almost every day). A total sum of points ranges from 0 to 21, where the higher score is associated with more serious anxiety. Regarding the severity category in this study, we followed the recommendations from the original version: not at all/normal (0 to 5 points), mild anxiety (5 to 9 points), moderate anxiety (10 to 14 points), and severe anxiety (15 to 21 points) [27,28].

Duke University Religion Index—DUREL as a measure for religiosity dimensions. This is a reliable measure of self-estimation of key religiosity dimensions. It comprises five items with responses rated on the Likert 1–5-point scale (1—does not apply to me at all; 5—applies to me completely). The questionnaire measures extrinsic religiosity through estimation of the organized religiosity (OR) and the non-organized religiosity (NOR). The other part of the questionnaire measures intrinsic religiosity (IR) [29,30].

Social support scale. This 8-item scale measures the extent of support in the form of “encouragement”, “useful information”, “direct assistance”, i.e., “offered things necessary to the subjects” etc., provided by people close to the subjects. Responses are rated from 1 (indicating minimal support) to 4 (indicating maximum support). The response “never” brings 1 point, “sometimes” brings 2 points, “often” brings 3 points, and “always” brings 4 points [31].

The World Health Organization WHOQOL-BREF quality of life assessment WHOQOL-BREF questionnaire. The version of the WHOQOL-100 questionnaire was created by the World Health Organization for the quality-of-life assessment [32,33,34]. The questionnaire contains 26 questions and each one is scored by the Likert scale from 1 (the worst) to 5 (the best) points. Following the transformation of points, which is conducted in 2 steps, the points for each domain are positioned on the 0–100 scale. From original results, a result for each domain of quality of life is calculated: physical health, psychic performance, social interaction, and environment, and additionally the overall satisfaction with quality of life and satisfaction with health are measured by two specific items which are considered separately. Results for each domain are calculated as a sum of results of each item. The points from each domain are transformed on the 0–100 scale in order to be compared with the original questionnaire and more legible presentation of results. A higher number of points means a higher quality of life in each of domains [33]. In this study, a validated Croatian version of the questionnaire was used [35].

These questionnaires were chosen in the present study because they were suitable for the aim of the study because they are internationally validated questionnaires already used and validated in Croatia in various studies.

We also tested several statements about self-treatment, based on previous qualitative work [22].

### 2.4. Independent Explanatory Variables:

Socio-demographic characteristics (age, gender, employment status, education level, marital status, financial status, place of residence);Self-treatment of pain. The questions were drawn from a previously conducted qualitative study of self-treatment of chronic non-malignant pain [22];Psychological factors: the degree of depression (measured by Center for Epidemiological Studies Depression Scale (CES-D) [26], and Anxiety (measured by Generalized Anxiety Disorder 7- item (GAD-7) scale) [27,28];Social factors. The social support was measured by a scale constructed for the purpose of Šverko et al. [31], which is an adapted version of the Abbey, Abramis, and Caplan scale [36];The outcome variable was the outcome of the treatment of chronic non-malignant pain. The outcome was dichotomous: the successful outcome (cured/improved) was defined as the control of pain achieved, with the pain intensity of 0–3 measured by NRS. The unsuccessful outcome (not cured) was defined as the pain intensity 4–10 on NRS even after one year of treatment or longer.

### 2.5. Statistical Analysis

The data were interpreted with a statistical significance level of at least 5%. The Kolmogorov-Smirnov test was used to assess the distribution of quantitative data. According to the findings, appropriate non-parametric statistical tests were used in the following analyses. Categorical variables were presented as frequencies and corresponding percentages while quantitative variables through medians and interquartile ranges (25th to 75th percentile).

Differences in the categorical variables were analyzed with the chi-square test while Fisher exact test or Fisher-Freeman-Halton exact test was used where cells contained less than 10 participants. Mann-Whitney U test was used to analyse differences between participants with successful chronic non-malignant pain management (NRS 0–3) and participants where the intensity of pain was not reduced despite treatment (NRS 4–10).

A binary logistic regression model was made to predict the poor outcome of the treatment of patients with chronic non-malignant pain with predictors that include the significant variables from previous univariate analyses.

*p* values below 0.05 were considered significant. Statistical software IBM SPSS Statistics, version 25.0 was used in all statistical procedures.

## 3. Results

### 3.1. Descriptive Statistics

#### Sample Parameters

A total of 180 participants were included in group 1 (participants where the intensity of pain was not reduced despite treatment (NRS 4–10), and 156 in group 2 (participants with successful chronic non-malignant pain management (NRS 0–3). Moreover, 24 patients were excluded from the analysis because they did not fully answer all the questions, i.e., they did not meet the set criteria.

Differences in categorical socio-demographic data between participants with successful chronic non-malignant pain management and participants where the intensity of pain was not reduced despite treatment (NRS 4–10) are shown in Table 1. There was a significant difference in educational level (NRS 0–3 group had greater number of higher educated participants, *p* = 0.003), working status (significantly higher number of retired people in NRS 4–10 group; *p* = 0.002), marriage status (*p* = 0.039), financial status (*p* = 0.002) and salary (*p* < 0.001). Some of these differences can be ascribed to significantly older age of NRS 4–10 group: 62.5 (54.0–71.8) years vs. 57.0 (46.3–66.0) years (*p* < 0.001. (Table 2)).

Differences in comorbidities between participants with successful chronic non-malignant pain management (NRS 0–3) and participants where the intensity of pain was not reduced despite treatment (NRS 4–10) are shown in Table 3. Participants where the intensity of pain was not reduced despite treatment (NRS 4–10) had a significantly higher prevalence of arthritis (*p* < 0.001), osteoporosis (*p* = 0.017), peripheral vascular disease (*p* = 0.029), diseases of the upper digestive system (*p* = 0.002), depression (*p* = 0.010), anxiety or panic disorder (*p* = 0.008), hearing and degenerative disorders (*p* = 0.022 and *p* < 0.001).

Table 3 Differences in comorbidities between the two groups of participants according to the outcome of chronic pain treatment.

Differences in general anxiety and depression scores between participants with successful chronic non-malignant pain management (NRS 0–3) and participants where the intensity of pain was not reduced despite treatment (NRS 4–10) are shown in Table 4.

In participants where the intensity of pain was not reduced despite treatment (NRS 4–10), there were higher level of anxiety (*p* < 0.001) and higher CES-D score (*p* < 0.001).

Differences in religiosity (The Duke University Religion Index-DUREL) and SFSS scores between participants with successful chronic non-malignant pain management (NRS 0–3) and participants where the intensity of pain was not reduced despite treatment (NRS 4–10) are shown in Table 5 (SFSS). There was no significant difference in the Durel score, but in participants where the intensity of pain was not reduced despite treatment (NRS 4–10), there was a significantly lower SFSS score (*p* = 0.035).

Differences in quality of life (The World Health Organization Quality of Life-Brief Version questionnaire, WHOQOL-BREF) between both groups are shown in Table 6. All of the quality-of-life domains were significantly poorer among participants where the intensity of pain was not reduced despite treatment (NRS 4–10) on the *p* < 0.001 level and in most cases (except social domain) with a median value below recommended quality of life values of 60.

Table 7 shows the differences in patient satisfaction between participants with successful chronic non-malignant pain management (NRS 0–3) and participants where the intensity of pain was not reduced despite treatment (NRS 4–10).

Participants with successful chronic non-malignant pain management (NRS 0–3) had significantly higher scores in PSQ18 Technical Quality (*p* = 0.005), PSQ18 Interpersonal Manner (*p* = 0.006) and PSQ18 Accessibility and Convenience (*p* = 0.047).

Differences in the self-treatment of pain practice between 2 groups of participants revealed a significantly higher proportion of positive answers on the claim “Self-treatment of pain was stimulated by a pharmacist” among NRS 4–10 group (22.8%) compared to NRS 0–3 group (12.2%) (*p* = 0.011). Moreover, there was a significantly higher proportion of positive answers on the claim that inaccessibility of doctors and therapies affects the decision to self-treatment of pain, with 31.8% positive answers in NRS 4–10 group and 21.2% positive answers in NRS 0–3 group (*p* = 0.028).

### 3.2. Inferential Statistics—Logistic Regression Analysis

Prediction of poor outcome of the treatment of patients with chronic non-malignant pain with the dimensions of pain (intensity, quality, localization), psychological factors (depression, anxiety), social factors (social support) and self-treatment is shown in Table 1. Binary logistic regression model is statistically significant (*p* < 0.001) and explains 64.5% of dependent variable variance. The model was statistically significant (*p* < 0.001), with r2 (Nagelkerke R Square) 64.5% and with 82.4% of correct classification of participants.

## 4. Discussion

The poor outcome of the treatment of chronic non-malignant pain in a multivariate binary logistic regression model is statistically significantly associated with the lower quality of life and higher depression level. The outcome of the treatment was not directly related to social support, but solitary life (without partner) was. The typical patient with the poor pain management outcome was retired, presented a depressive mood, and had pain that disturbed general activity and sleeping. Following bivariate analysis, several predictors showed a significant prediction of belonging to a group with unsuccessful treatment outcome (Table 8).

Among socio-demographic characteristics, the most significant predictor was retired compared to unemployed, followed by living alone. Although women prevailed in both study groups, gender still did not appear to be a predictor of unsuccessful outcome. It is assumed that women more frequently report their pain and differently react to pain, as described by Wijnhoven HA et al. [37]. Regarding the age of subjects, those in the group with unsuccessful outcome were somewhat older compared to those with successful pain treatment outcome, but age still did not appear to be a significant predictor. The age in this study can speak in favor of results indicating that retired subjects were predictors of unsuccessful treatment outcome.

In the study of Breivik et al., elderly persons who suffered from chronic pain were more represented in most of the countries studied, for example, in Germany, the Nordic countries, the Netherlands, France and Spain, while in Israel, Poland and Italy younger age groups suffered more from chronic non-malignant pain [1]. In a large study carried out in 43 countries, Stubbs et al. demonstrated significant association of chronic pain with older age, female gender, lower educational level, and urban environment [38]. Azevedo et al. also found out the same [39]. Several studies demonstrated similar risk factors associated with chronic pain, such as low level of education, low family income, manual work, and being single (those living without partners) [40,41,42].

Studies demonstrated that social support is associated with better general health conditions and a higher subjective sense of well-being because it provides people with a feeling of belonging and support [43] and it is directly related to satisfaction with life [44]. It is more likely that people with closer relationships will: be more included in social activities; have better appetite; have healthier behavior and pay more attention to their health [45]. In our study, we found out the association of higher social support and favorable outcome of treatment of pain in bivariate statistics, but social support was not a significant predictor in the multivariate model.

Holtzman et al., in their study, show that social support has an effect on chronic pain intensity, and various sources of support can encourage the person to use the facing strategy [20]. Other studies also suggest that the combination of emotional support of family or other important persons, in combination with informational physician’s support, can help in the process of acceptance in persons with chronic pain [46]. Encouraged and emotionally supported patients have a higher probability of participation in every day, social, professional, and extracurricular activities [47,48,49,50,51]. Still, the social support changes over time and becomes limited [52] due to the chronic nature of pain, which is persistent, long-lasting, and often silent [53,54].

Among psychological factors, the significant predictor in the model of our study was the higher CES-D Score. These results are supported by other studies which indicated that depression and anxiety, as the most common disorders in a general population [55,56] are frequently present in patients with chronic pain as well [2]. According to some authors, depression becomes a key factor and predictor of painful symptoms such as musculoskeletal pain [57,58]. Beyer believes that psychosocial factors have higher impact on pain chronification than somatic problems and that depression and anxiety are the main triggers in pain chronification [59]. Other studies also revealed the association of higher intensity pain with depression [7] and that concomitantly present depression complicates the treatment of pain and reduces the success of treatment [7,8,9,10], which is in line with our results.

In our model level of anxiety was not a predictor of the treatment outcome, but we found an association between anxiety and worse outcomes in bivariate statistics.

Lower WHOQOL-BREF PHYS) also showed the association with unsuccessful treatment. Since this study was cross-sectional it was difficult to determine whether the lower WHOQOL-BREF PHYS was the cause or the consequence of unsuccessful treatment outcome, which is at the same time one of the disadvantages of cross-sectional studies. However, the bivariate analysis revealed that all of the WHOQOL-BREF domains were significantly lower in the group with the unsuccessful outcome of pain treatment, while in the model, logistic regression showed that only the physical domain was a predictor of an unsuccessful outcome. Other studies also support the conclusion that chronic pain is associated with a negative impact on the quality of life, both physical and mental [60] and results in its aggravation [61]. The chronic pain is a complex experience. Except that it is disturbing for an individual because of its long-lasting nature, it has also psychological, social, and economic consequences. It causes problems with walking, house chores and engaging in even simple activities such as sitting or standing [1,62,63]. In addition, chronic pain impacts social interaction as well, because it constrains patient’s activities and social contacts. Such an interpenetration of all aspects of life caused by chronic pain eventually results in the patient’s lower quality of life [64].

In addition to other factors, the outcome of chronic pain treatment can be influenced by the factor of patient satisfaction with the support of healthcare providers and with function of healthcare services. Among seven domains assessed, we gathered statistically significant differences between two study groups in three domains. The results obtained indicated that participants with successful chronic non-malignant pain management had significantly higher scores in PSQ18 Technical Quality, Interpersonal Manner and Accessibility and Convenience. It can be concluded that the patient’s satisfaction with health services was of great importance, where it is important to identify weaknesses in the healthcare system by “patient’s eyes”, as it allows weaknesses identified by patients to accomplish results in its improvement [65]. Positive or negative patient’s assessments of certain dimensions are of vital importance for monitoring the quality of care in medical institutions [66]. The patient satisfaction, as a predictor of a more favorable treatment outcome, may be the result of better acceptance of medical advice and treatment, use of services and improvement of relations between the physician and patient [67].

Significant predictor of an unsuccessful outcome of chronic, non-malignant pain treatment was the self-treatment from the inaccessibility of doctors and therapies. Similar results were gathered in the first, qualitative part of the study, where the inaccessibility of doctors and long waiting time for the treatment were the most significant factors associated with the decision on self-treatment, which has been confirmed by other studies as well [68,69]. According to Loese and Melzac, in spite of various treatment options, chronic pain will probably persist even after the end of treatment and will be understood as a condition that has not been cured [70]. This indicates that, in many cases, patients have to treat their pain themselves and on a daily basis [71] and, according to Barlow, take actions to identify, treat and manage, and adopt behaviors that influence their health [72]. The biopsychosocial model of chronic pain treatment implies the program of multidisciplinary approach which guides the patient to adopt his/her diagnosis and learn how to live with chronic pain [73,74]. According to this model, psychological and behavioral factors (self-assessment of quality of life) and social factors (educational level, medical care availability), in addition to biological ones, have also a high impact on disease outcome i.e., disease resolution [75].

## 5. Conclusions

It can be said that the typical patient with an unsuccessful outcome of pain management in our study was retired, living alone, with depressive behavior and with pain that disturbed general activity (physical aspect of QoL) and sleeping. Since the outcome of chronic, non-malignant pain treatment is, in addition to medical factors, influenced by many other, biopsychosocial factors, the approach to treatment should be adjusted to those diversities. Based on the results of our study, it can be concluded that the principle of treatment of patients with chronic, non-malignant pain should take into account a biopsychosocial approach with individually adjusted procedures.

## 6. Limitations of the Study

Since this study was cross-sectional, it is not possible to determine whether some factors were the cause or the consequence of unsuccessful treatment outcome, which is at the same time one of the disadvantages of cross-sectional studies. For instance, logistic regression has shown that retirement status is a strong predictor of the outcome when treating chronic, non-malignant pain. However, it is debatable whether the patients in retirement experience stronger pain or if the chronic pain has driven patients into early retirement.

The weaknesses of this study are not only methodological but also statistical. For testing differences in quantitative variables, we were forced to use the Mann-Whitney U (MWU) test because the Kolmogorov-Smirnov (KS) test has shown the non-normal distribution in measured variables. The MWU test is generally less powerful than its parametric counterpart [76], so there is an increased probability of false-negative results in some variables. For non-normal quantitative data, a transformation could be used to obtain a more Gaussian distribution, however, since the interpretation of transformed data is usually less intuitive, such procedures were not used.

## Figures and Tables

**Table 1 ijerph-19-02881-t001:** Differences in sociodemographic data between the two groups of participants according to the outcome of chronic pain treatment: (χ^2^) test.

Socio-Demographic Data		Groups				*p*
		NRS 4–10		NRS 0–3		
		N	%	N	%	
Gender	Male	26	14.44%	35	22.44%	0.058
	Female	154	85.56%	121	77.56%	
**Education**	Elementary school	38	21.11%	18	11.54%	**0.009**
	University or college degree	102	56.67%	84	53.85%	
	Master’s or Ph.D. degree	40	22.22%	54	34.62%	
Place of living *	Village	25	13.89%	16	10.26%	0.697
	Small town (<5000 inhabitans)	12	6.67%	9	5.77%	
	Bigger town (5000–50,000 inhabitans)	19	10.56%	15	9.62%	
	City (>50,000 inhabitans)	124	68.89%	116	74.36%	
**Working status ***	Unemployed	10	5.56%	18	11.54%	**<0.001**
	Employed	59	32.78%	76	48.72%	
	Retired	111	**61.67%**	62	**39.74%**	
**Marriage ***	Living alone	77	42.78%	49	31.41%	**0.032**
	Living with partner	103	57.22%	107	68.59%	
**Financial status ***	Below average	63	**35.00%**	32	**20.51%**	**0.002**
	Average	115	63.89%	116	74.36%	
	Above average	2	1.11%	8	5.13%	
**Salary ***	<3000 kn	101	56.11%	49	31.41%	**<0.001**
	3000–6000 kn	68	**37.78%**	80	**51.28%**	
	6000–10,000 kn	11	**6.11%**	22	**14.10%**	
	>10,000 kn	0	0.00%	5	3.21%	

* Fisher-Freeman-Halton’s test, values in bold are statistically significant.

**Table 2 ijerph-19-02881-t002:** Age differences between the two groups of participants according to the outcome of chronic pain treatment: Mann-Whitney U test.

Groups		N	Minimum	Maximum	Percentiles			*p*
					25th	50th (Median)	75th	
**Age (years)**	NRS 4–10	180	28.00	88.00	54.00	**62.50**	71.75	**<0.001**
	NRS 0–3	156	20.00	83.00	46.25	**57.00**	66.00	

Values in bold are statistically significant.

**Table 3 ijerph-19-02881-t003:** Functional Comorbidity Index (FCI).

Groups		N	Minimum	Maximum	Percentiles			*p*
					25th	50th (Median)	75th	
The Functional Comorbidity Index (FCI)	NRS 0–3	156	0.00	9.00	1.00	2.00	3.00	**<0.001**
	NRS 4–10	180	0.00	10.00	2.00	3.00	5.00	

Values in bold are statistically significant.

**Table 4 ijerph-19-02881-t004:** Differences in general anxiety and depression between the two groups of participants according to the outcome of chronic pain treatment: Mann-Whitney U test.

		N	Minimum	Maximum	Percentiles			*p*
					25th	50th (Median)	75th	
**GAD-7 score**	NRS 4–10	180	0.00	21.00	4.00	**7.00**	12.00	**<0.001**
	NRS 0–3	156	0.00	21.00	1.00	**3.00**	6.75	
**CES-D score**	NRS 4–10	180	3.00	51.00	18.00	**25.00**	33.75	<0.001
	NRS 0–3	156	3.00	47.00	8.00	**14.00**	21.75	

Values in bold are statistically significant.

**Table 5 ijerph-19-02881-t005:** Differences in religiosity (The Duke University Religion Index-DUREL) and social factors social support (SFSS) scores between the two groups of participants according to the outcome of chronic pain treatment: Mann-Whitney U test.

		N	Minimum	Maximum	Percentiles			*p*
					25th	50th (Median)	75th	
DUREL score	NRS 4–10	180	5.00	27.00	13.00	17.00	21.75	0.583
	NRS 0–3	156	5.00	26.00	13.00	17.00	21.00	
SFSS	NRS 4–10	180	9.00	28.00	18.00	22.00	26.00	**0.035**
	NRS 0–3	156	13.00	28.00	20.25	23.00	25.00	

Values in bold are statistically significant.

**Table 6 ijerph-19-02881-t006:** Differences in quality of life between the two groups of participants according to the outcome of chronic pain treatment: Mann-Whitney U test.

		N	Minimum	Maximum	Percentiles			*p*
					25th	50th (Median)	75th	
**WHOQOL-BREF PHYS**	NRS 4–10	180	3.57	71.43	32.14	**39.29**	50.00	<0.001
	NRS 0–3	156	25.00	92.86	43.75	**57.14**	67.86	
**WHOQOL-BREF PSYCH**	NRS 4–10	180	16.67	100.00	45.83	**54.17**	70.83	<0.001
	NRS 0–3	156	33.33	95.83	58.33	**70.83**	79.17	
**WHOQOL-BREF SOCIAL**	NRS 4–10	180	8.33	100.00	50.00	**66.67**	75.00	<0.001
	NRS 0–3	156	25.00	100.00	58.33	**75.00**	83.33	
**WHOQOL-BREF ENVIR**	NRS 4–10	180	15.63	93.75	46.88	**56.25**	65.63	<0.001
	NRS 0–3	156	31.25	90.63	56.25	**65.63**	74.22	

Values in bold are statistically significant.

**Table 7 ijerph-19-02881-t007:** Differences in patient satisfaction (PSQ18) between the two groups of participants according to the outcome of chronic pain treatment.

		N	Minimum	Maximum	Percentiles			*p*
					25th	50th (Median)	75th	
PSQ18 General Satisfaction	NRS 4–10	180	1.00	5.00	2.00	3.00	3.50	0.539
	NRS 0–3	156	1.00	5.00	2.50	3.00	3.50	
**PSQ18 Technical Quality**	NRS 4–10	180	1.50	5.00	2.75	**3.25**	3.75	**0.005**
	NRS 0–3	156	1.50	4.50	3.00	**3.50**	3.75	
**PSQ18 Interpersonal Manner**	NRS 4–10	180	1.00	5.00	3.00	**3.50**	4.00	**0.006**
	NRS 0–3	156	2.00	5.00	3.50	**4.00**	4.38	
PSQ18 Communication	NRS 4–10	180	1.00	5.00	2.50	3.50	4.00	0.099
	NRS 0–3	156	1.50	4.50	3.00	3.50	4.00	
PSQ18 Financial Aspects	NRS 4–10	180	1.00	5.00	2.00	3.00	4.00	0.269
	NRS 0–3	156	1.00	5.00	2.50	3.00	4.00	
PSQ18 Time Spent with Doctor	NRS 4–10	180	1.00	5.00	2.50	3.00	4.00	0.447
	NRS 0–3	156	1.00	5.00	2.50	3.00	4.00	
**PSQ18 Accessibility and Convenience**	NRS 4–10	180	1.00	5.00	2.00	**2.50**	3.25	**0.047**
	NRS 0–3	156	1.00	5.00	2.25	**2.75**	3.50	

Values in bold are statistically significant.

**Table 8 ijerph-19-02881-t008:** Prediction of poor outcome of the treatment of patients with chronic non-malignant pain with the dimensions of pain (intensity, quality, localization), psychological factors (depression, anxiety), social factors (social support) and self-treatment: binary logistic regression.

	OR	95% CI		*p*
		Lower	Upper	
Age (years)	0.99	0.96	1.03	0.753
Working status: unemployed				0.089
Working status: employed	1.58	0.48	5.29	0.454
**Working status: retired**	**3.73**	1.07	12.97	**0.039**
Education	0.83	0.56	1.24	0.375
**Living alone**	**2.16**	1.03	4.53	**0.043**
Financial status: below average				0.568
Financial status: average	0.73	0.32	1.67	0.455
Financial status: above average	0.25	0.01	4.82	0.357
GAD-7 score	0.91	0.83	1.00	0.057
**CES-D score**	**1.08**	1.02	1.14	**0.009**
SFSS	1.04	0.95	1.15	0.395
**WHOQOL-BREF PHYS**	**0.95**	0.91	0.99	**0.009**
WHOQOL-BREF PSYCH	1.03	1.00	1.07	0.067
WHOQOL-BREF ENVIR	1.00	0.97	1.04	0.845
Self-treatment of pain was stimulated by a pharmacist	0.80	0.34	1.89	0.614
**Self-treatment: inaccessibility of doctors and therapies**	**2.89**	1.30	6.44	**0.009**
PSQ18 Technical Quality	1.33	0.64	2.77	0.440
PSQ18 Interpersonal Manner	0.67	0.40	1.13	0.132
PSQ18 Accessibility and Convenience	1.29	0.76	2.19	0.345

Values in bold are statistically significant.

## Data Availability

The datasets generated and analyzed for this study can be requested from the correspondent author. The data are not publicly available due to policy of institutions which gave ethical approval to the study.

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
