# Peer review of "Non-Medical Factors Associated with the Outcome of Treatment of Chronic Non-Malignant Pain: A Cross-Sectional Study"

_ijerph, 2022, doi:10.3390/ijerph19052881_

Round 1

Reviewer 1 Report

An interesting article that relates aspects of lifestyle and pain management and perception. 
Some recommendations to the authors:
Improve the English of the article.
The conclusions of the abstract should be in line with those of the text.
In the introduction there are paragraphs such as the one in lines 50-65 that should be written in a clearer way. There are recurring phrases. In line 72 there are changes in the citation form. 
Although line 386 states that informed consent has been given, this should be reflected in the method, as well as the ethical code applied.
Clarify the validation of the instruments.
Well-described Independent explanatory variables and Statistical analysis        
Results well presented, I consider it unnecessary to use different colours. In any case, use of bold and in a low proportion.
Excess of tables, some of them probably unnecessary.
In paragraph 324-339 some colloquial phrases that should be changed.
In the limitations, clarify what these limitations may imply.
References should be adapted to the format of the journal. Some of the bibliographical references do not include direct access to the document.   

Author Response

Dear Editor and the Reviewer,

We thank you for the comments and advice on how to improve the paper. We answered all the comments, and we hope that we could explain and correct all your suggested drawbacks of the paper.

Kind regards,

Irena Kovačević

Reviewer 2 Report

This is an interesting study clarifying the non-medical factors associated with chronic pain states. I have no major concerns, but few comments that I think the authors would like to consider to make the manuscript clearer.

Page 4, row 148, please check spelling: “The date were interpreted with a significance level of at least 5 %. “

The title says “non-medical factors associated with”, I think that depression is a medical factor?

Methods:

I believe that the used indexes are valid, but there are so many of them that at least I am not familiar with all of them. It would be helpful to introduce the main points of each index and questionnaire. I know that these could be found in the references, but the readability suffers too much if I need to search the basic information from 10 references while I am reading an interesting article. Also, the references do not provide the information why these questionnaires were chosen in the present study.

Results:

The authors have made an effort and collected a large set of data, and they also deserve a credit of highlighting the significant findings in the tables with color. However, the large amount of data presented in 9 tables also makes the manuscript confusing. Narrative descriptions of the results are quite shallow, and it is quite a “jungle” to find the essential information from the tables. In my opinion, the findings should be highlighted more clearly and the tables should not be the results, they should support them. As a conclusion, I suggest that the subheadings describe the main findings, and the narrative results describe significant findings in a qualitative and quantitative manner.

Discussion:

The structure of the discussion is a bit odd, and the first sentence provides no useful information. I suggest that the first paragraph summarizes the main findings of the present study to make it more clear.

The discussion is interesting, but there is quite a little discussion about the findings of the present study. For an example, retirement and living alone were found to be significant predictors whereas gender did not make difference (page 9, rows 256–259). After that, the discussion deals with the gender (non-significant predictor) to row 264. Age and living alone, the significant predictors are carefully mentioned later (rows 267 and 277, respectively). This regards few of the later paragraphs as well. The discussion should focus more on the present findings.

The authors have compared their results on the earlier literature with a good analytic touch. However, when I read a sentence that tells me “contrary to earlier literature”, the next thing that I am waiting with great interest is “why”? Could the authors discuss about the reasons to different outcomes?

Exact results (OR, CI etc.) belong to the results section, this is the info that I meant in the results comment above. Here you may discuss about the findings without p-values etc.

Limitations of the study section is appreciated. The number on row 371 (MWU test is generally less powerful than its parametric counterpart (258) so there is an increased probability of false-negative results in some variables.) should probably be a reference?

Author Response

(The authors gave the same response as above.)

Reviewer 3 Report

ijerph-1602174

Non-Medical Factors Associated With The Outcome of 2 Treatment of Chronic Non-Malignant Pain: A Cross - Sectional Study

Introduction

Chronic pain and chronic non-malignant pain are referred to in the introduction I recommend just focusing on one and being concise.

The second paragraph has multiple definitions of chronic pain. I would recommend only having one.

Reference is needed to support the statement “Literature review showed insufficient data in this field especially in Croatian settings.”

Methods

In the section “type of research”, Was the questionnaire used in the study and validated questionnaire? I see that such details are mentioned later. I recommend collating similar content together. The details about the questionnaire at the start of the methods should go under the subheading of “instrument”.

To confirm, in the section of participants, those included in the study were based on if the patient had recovered or not? This is an important aspect of the eligibility criteria otherwise number five outcome independent explanatory variable on page 3 becomes redundant.

Why were the patients in the first group who had successful treatment still having follow-up? Were there any specific exclusion criteria?

I assume the questionnaires were answered by participants at their follow-up consultation? Could this be at any follow-up time or was it standardised e.g. on the second appointment?

To confirm in section about the “Independent explanatory variables”, doesSelf-treatment of pain” mean participants support of treatment they used to manage their pain or is it their pain intensity? If it’s the former I’m surprise that pain intensity was not captured.

I am unclear on the relationship between date and statistical significance in the statement “The date were interpreted with a significance level of at least 5 %.”

Results

Table 1

I am unsure why some text in Table 1 is in red colour. This also occurs in Table 4, 6-9. If it is related to statistical significance the use of colour, particularly red (due to red/green colour blindness) is not the best way to indicate this.

Salary is misspelt

There are undefined abbreviations e.g SSS, inh etc

Table 3 details the functional comorbidity index but in the text above the table there is reference to various comorbidities (“Participants where the intensity of pain was not reduced despite treatment had significantly 185 higher prevalence of arthritis, osteoporosis, peripheral vascular disease, diseases of the 186 upper digestive system, depression, anxiety, or panic disorder, hearing and degenerative 187 disorders”) which we cannot be deduced from this. This the text the authors view or it is a finding? If it is the latter, the proportion of participants with each comorbidity should be stated.

Author Response

(The authors gave the same response as above.)

Round 2

Reviewer 1 Report

Thanks for the great effort made to adapt the article to the suggested changes.

Some recommendations to the authors:

The abstract should include some reference to limitations.

Concerning the ethical considerations in line 97, I don't consider it sufficient to refer to the fact that patients were asked. This kind of research should have been submitted to an ethics committee and this should be made clear in the text.

Author Response

Thank you for your effort and your time as well as for the suggestions to improve this manuscript. We tried to correct it according to the instructions.

Kind regards,

Irena Kovačević
